# The Corticosterone–Glucocorticoid Receptor–AP1/CREB Axis Inhibits the Luteinizing Hormone Receptor Expression in Mouse Granulosa Cells

**DOI:** 10.3390/ijms232012454

**Published:** 2022-10-18

**Authors:** Xuan Zhang, Yinghui Wei, Xiaoxuan Li, Chengyu Li, Liangliang Zhang, Zhaojun Liu, Yan Cao, Weijian Li, Xiying Zhang, Jiaqing Zhang, Ming Shen, Honglin Liu

**Affiliations:** 1College of Animal Science and Technology, Nanjing Agricultural University, Nanjing 210095, China; 2Hangzhou Academy of Agricultural Sciences, Hangzhou 310024, China; 3Institute of Animal Husbandry and Veterinary Science, Henan Academy of Agricultural Sciences, Zhengzhou 450002, China

**Keywords:** corticosterone, granulosa cells, luteinizing hormone receptor, glucocorticoid receptor

## Abstract

Under stress conditions, luteinizing hormone (LH)-mediated ovulation is inhibited, resulting in insufficient oocyte production and excretion during follicular development. When the body is stressed, a large amount of corticosterone (CORT) is generated, which will lead to a disorder of the body’s endocrine system and damage to the body. Our previous work showed that CORT can block follicular development in mice. Since LH acts through binding with the luteinizing hormone receptor (*Lhcgr*), the present study aimed to investigate whether and how corticosterone (CORT) influences *Lhcgr* expression in mouse ovarian granulosa cells (GCs). For this purpose, three-week-old ICR female mice were injected intraperitoneally with pregnant mare serum gonadotropin (PMSG). In addition, the treatment group was injected with CORT (1 mg/mouse) at intervals of 8 h and the control group was injected with the same volume of methyl sulfoxide (DMSO). GCs were collected at 24 h, 48 h, and 55 h after PMSG injection. For in vitro experiments, the mouse GCs obtained from healthy follicles were treated with CORT alone, or together with inhibitors against the glucocorticoid receptor (*Nr3c1*). The results showed that the CORT caused a downregulation of *Lhcgr* expression in GCs, which was accompanied by impaired cell viability. Moreover, the effect of the CORT was mediated by binding to its receptor (*Nr3c1*) in GCs. Further investigation revealed that *Nr3c1* might regulate the transcription of *Lhcgr* through inhibiting the expression of *Lhcgr* transcription factors, including AP1 and Creb. Taken together, our findings suggested a possible mechanism of CORT-induced anovulation involving the inhibition of *Lhcgr* expression in GCs by the CORT–*Nr3c1*–AP1/Creb axis.

## 1. Introduction

With the development of intensive production in animal husbandry, to maximize the production level of livestock and poultry and to improve economic benefits, some production processes and technical measures are adopted which often deviate from the natural environmental conditions to which livestock and poultry are adapted. For instance, under the conditions of intensive farming, poultry and livestock are inevitably affected by various environmental factors and thus show a state of stress, which reduces the reproductive performance of livestock and poultry and ultimately affects the quality and yield of livestock products [1]. In addition, stress has been shown to disrupt ovulatory cyclicity and induce a series of ovarian diseases, resulting in reproductive disorders and reproductive health challenges in humans, such as polycystic ovary syndrome (PCOS) [2]. Therefore, it is of great importance in animal production to explore the inhibitory mechanism of stress on animal reproductive performance and find the molecular target of stress.

Due to sudden changes in the environment, such as temperatures that are too high or too low, noise, excessive light, and shock, a series of environmental stimuli produce physical or psychological stress in the animal body. Stress is accompanied by neurocrine and endocrine changes, especially in the adrenal medulla and cortex [3]. Upon stress response, the hypothalamus–pituitary–adrenal (HPA) axis is activated, and glucocorticoid levels increase rapidly. Cortisol (in non-rodent mammals) and corticosterone (in rodents) is the main component of glucocorticoids (referred to herein as CORT). In mammals, corticosterone is mainly produced by the adrenal cortex and is one of the central response factors of the stress response. The ovaries are particularly sensitive to hormones during the period from growth and development to before and after ovulation. When rodents are exposed to stress, the level of corticosterone will increase rapidly in the serum. Because the concentration of corticosterone is higher than that of cortisol, corticosterone is regarded as the main stress index [4].

Luteinizing hormone (LH) is a glycoprotein hormone secreted by the anterior pituitary gland in response to the hypothalamic gonadotropin-releasing hormone [5]. The LH receptor (*Lhcgr*) is a key hormone receptor that induces ovulation and luteal formation. After binding with LH in granulosa cells (GCs), *Lhcgr* induces the growth of preovulatory follicles. The expression of *Lhcgr* undergoes dynamic changes in the normal ovarian cycle. Initially, *Lhcgr* is expressed in follicular membrane cells before secondary follicles [6]. With the synergistic effect of FSH, estradiol, and other paracrine factors, *Lhcgr* expression levels increase in GCs during follicular growth and reach a maximum level before ovulation [7,8,9]. The combining of LH with *Lhcgr* in GCs activates the AC/cAMP/PKA, PI3K/AKT, and RAS signaling, which are essential for ovulation [10]. LH binds to its G protein-coupled receptor *Lhcgr* and initiates the classic AC/cAMP/PKA pathway, resulting in phosphorylation and the activation of the transcription factor Creb. The activation of steroid synthase-related genes through this signaling pathway increases the expression of synthase genes, such as aromatase [11]. Aromatase is a key gene for the synthesis of steroid hormones in the follicles and is highly expressed in ovarian granulosa cells. The activated Creb can act as a transcription enhancer to stimulate gene transcription and upregulate *Lhcgr* expression. The promoter of the *Lhcgr* gene also includes a binding site for AP1. It has been reported that AP1 binding sites are essential for activating cAMP-induced *Lhcgr* transcription [5]. The transcription of *Lhcgr* is regulated by a variety of transcription factors, such as Creb and AP1. They are two well-described transcription factors which serve important roles in ovarian functions [12]. The AP1 transcription factor usually expresses its activity as a homodimer or heterodimer of c-Jun and c-Fos. Creb is a ubiquitous transcription factor that controls cell proliferation and survival. When the *Lhcgr* in GCs is knocked out, even with high-dose FSH stimulation, follicular development cannot be restored during the follicular phase. Therefore, the expression of *Lhcgr* in ovarian GCs is essential for follicle maturation and ovulation [13,14].

Conditions such as stresses, diseases (such as Cushing’s syndrome), psychological disorders, and the long-term administration of synthetic glucocorticoids could promote the excessive production of CORT via activating the hypothalamus–pituitary–adrenal (HPA) axis [15,16]. The overproduction of glucocorticoids is known to disturb the endocrine system, suppress the immune system, damage the reproductive system, and cause reproductive disorders [17,18,19]. Glucocorticoids inhibit the HPO axis at the level of the hypothalamus, pituitary gland, ovary, and uterus [19,20]. When CORT was increased sharply in rats, changes in their sexual behavior were observed, and the secretion of some hormones, such as plasma ACTH, CORT, and PRL, increased significantly. On the other hand, the secretion of hormones such as FSH decreased [21]. There is evidence that the activation of the HPA axis induces a surge of CORT, which then reduces the pulse secretion of GnRH/LH, leading to impaired ovulation due to the lack of LH support [22]. During the follicular phase in ewes, high concentrations of CORT inhibit the frequency of LH secretion, resulting in low fertility [18]. Early laboratory experiments have shown that mice in estrus suffer from decreased ovulation due to increased CORT levels in the body [23]. Moreover, there is evidence showing that *Lhcgr* expression is reduced during CORT overproduction [24]. A previous study has shown that cortisol injection in female mice significantly increase FasL expression in ovaries and impair oocyte competence by activating the Fas signaling and triggering apoptosis in MGCs, CCs, and oocytes [25]. In fact, the role of glucocorticoids is mediated by classic glucocorticoid receptors (*Nr3c1*) [26,27,28]. *Nr3c1* belongs to the nuclear receptor family and acts as a transcription regulator that can affect the function and recruitment of transcription factors [29,30]. After combining with CORT, *Nr3c1* translocates to the nucleus, interacting with GREs and regulating the expression of target genes [26,31]. However, whether CORT can regulate the expression of *Lhcgr* in ovarian GCs through *Nr3c1* remains to be studied.

In response to harmful environmental conditions, the reproduction process of animals is accompanied by a surge of CORT. Our previous experiments have confirmed that continuous injections of CORT can be used to build a mouse stress model, which negatively affects the ovulation process [23]. Considering the fact that LH-induced ovulation requires the binding of LH with *Lhcgr* in ovarian GCs, we asked whether CORT might influence the expression of *Lhcgr* in GCs. To test this hypothesis, we determined *Lhcgr* expression in cultured GCs with CORT treatment in GCs collected from mice receiving continuous injections of CORT. Both in vivo and in vitro results showed that CORT causes a downregulation of *Lhcgr* expression in GCs. We next explored the mechanism of CORT-mediated suppression of *Lhcgr* expression, and demonstrated that CORT acts through *Nr3c1* to inhibit the expression of Creb and AP-1, resulting in the blocking of *Lhcgr* transcription in GCs. These findings might lay a theoretical foundation for developing protocols for use in alleviating the adverse effects of various levels of CORT on animal reproduction. Moreover, these results and conclusions provide therapeutic ideas and targets for the improvement of human and animal reproductive health.

## 2. Results

### 2.1. CORT Inhibits LH Receptor Expression in the Ovarian Granulosa Cells of Mice

We first detected whether CORT might affect *Lhcgr* mRNA expression in mouse ovarian sections. The results of the immunofluorescence assays show that the protein level of *Lhcgr* had significantly decreased in ovaries collected from mice injected with CORT in a time-dependent manner (Figure 1A,B). Consistent with this, as shown in Figure 1C, the mRNA levels of *Lhcgr* in the GCs of the control group were significant at 55 h. In contrast, *Lhcgr* expression was inhibited by CORT injection during follicular growth. In particular, at 55 h after PMSG injection, both the protein and mRNA levels of *Lhcgr* in the CORT group were significantly lower than those of the control group.

We next determined the effects of CORT in in vitro cultured GCs. Consistent with the in vivo results, CORT inhibited the *Lhcgr* mRNA levels in a dose-dependent manner (Figure 1D). GCs treated with 200 μM of CORT also showed a significant decrease in *Lhcgr* protein level (Figure 1E). Moreover, a loss of cell viability was observed in GCs with CORT treatment (Figure 1F).

### 2.2. The CORT-Mediated Inhibition of Lhcgr Expression Is Dependent on the Activation of the Glucocorticoid Receptor (Nr3c1) in Mouse Ovarian GCs

The physiological functions of CORT are induced by activating downstream signaling transduction through binding with the glucocorticoid receptor (*Nr3c1*) [32]. Therefore, we evaluated whether *Nr3c1* is involved in the CORT-mediated regulation of *Lhcgr* expression in mouse GCs. Using immunofluorescence, we first determined the localization and level of *Nr3c1*, a specific receptor of CORT, in mouse ovary sections. As shown in Figure 2A the protein level of *Nr3c1* in the CORT group was higher than that in the control group at 48 h and 55 h after PMSG injection. In accordance with the in vivo results, CORT treatment of primary GCs in vitro also showed a marked elevation of *Nr3c1* expression (Figure 2D,E).

To further clarify whether CORT regulates *Lhcgr* mRNA expression by activating the *Nr3c1* pathway, we treated GCs with the *Nr3c1* inhibitor RU486. The results of Figure 3A,B show that RU486 eliminated the inhibitory effects of CORT on *Lhcgr* expression at both protein and transcription levels. To improve the specificity for inhibiting *Nr3c1* mRNA expression, we knocked down *Nr3c1* expression using two paralleled *Nr3c1* siRNAs (See Figure 3). Consistent with the results from the RU486 treatment, the *Nr3c1* siRNAs also restored *Lhcgr* mRNA and protein expression in the CORT-treated GCs (Figure 3C,D).

### 2.3. CORT Inhibits the Transcriptional Activity of Lhcgr Transcription Factors in GCs

The active form of AP1 consists of c-Jun and c-Fos, which can dimerize to form the AP1 complex. To further elucidate the underlying mechanism of CORT-mediated inhibition of *Lhcgr* expression in GCs, we collected GCs at 55 h after PMSG injection and analyzed the expression of *Creb1, Jun*, and *Fos* using western blot and qRT-PCR. As shown in Figure 4A–C, CORT reduced the ratio of phosphorylated Creb (Ser133) to total Creb, and downregulated the expression of AP1 subunits (including c-Fos and c-Jun).

We next determined the expression of AP1 subunits and Creb in cultured GCs. Consistent with the in vivo results, the ratio of p-Creb to Creb, as well as the protein levels of c-Jun and c-Fos, were significantly decreased after CORT treatment in cultured GCs in vitro (Figure 5A–C). To assess whether *Lhcgr* downregulation is related to the CORT-induced suppression of Creb and AP1, we used 666-15 (Creb-specific inhibitor) and T-5224 (AP1-specific inhibitor) to inhibit Creb and AP1 activity, respectively. As shown in Figure 5D–E, *Lhcgr* protein levels were downregulated when the inhibitors were used.

### 2.4. The Expression of AP1 and Creb Is Regulated by Nr3c1

To further clarify the signaling cascades linking CORT, *Nr3c1*, and *Lhcgr* expression, we treated mouse GCs with RU486 and analyzed the expression of c-Jun, c-Fos, and Creb using qRT-PCR and western blot. As shown in Figure 6A–E, RU486 restored c-Jun and c-Fos expression and the ratio of p-Creb to Creb in the CORT-treated GCs. Consistent with the results from the RU486 treatment, *Nr3c1* siRNAs also restored the protein levels of c-Jun and c-Fos and the ratio of p-Creb to Creb in the CORT-treated GCs (Figure 6F,G). To test whether Creb and AP1 would directly activate the expression of *Lhcgr* at the promoter level, fragments of the *Lhcgr* promoter containing the putative Creb/c-Jun binding sites or mutated Creb/c-Jun binding sites were cloned upstream of a luciferase reporter gene as pGL3-*Lhcgr* (WT), pGL3-*Lhcgr* (M1, mutated Creb binding site), and pGL3-*Lhcgr* (M2, mutated c-Jun binding site), respectively (Figure 6). These constructs were then transfected into GCs. As shown in Figure 6H,I, both the CORT treatment and binding site mutations markedly reduced the responsiveness of the luciferase reporter to Creb and c-Jun (Figure 6H,I). These data suggest that CORT might act through Creb and AP1 to inhibit *Lhcgr* transcription in GCs.

## 3. Discussion

Environmental stresses have a negative impact on ovarian development and ovulation in female mammals. Under stress or certain physiopathological conditions, high levels of plasmic CORT are generated through the HPA axis, leading to a decreased synthesis and release of reproductive hormones, which in turn inhibit follicular development and ovulation in female animals [33,34,35]. Studies have shown that when glucocorticoid exceeds the normal level, it will work through *Nr3c1*, which acts as a nuclear transcription factor to regulate the transcription of target genes. As the main component of glucocorticoids, CORT was recognized to negatively affect the ovulation process in our previous studies, although the underlying mechanism remains unclear. Here, we demonstrated that continuous intraperitoneal injection of CORT on mice downregulates *Lhcgr* expression in ovarian GCs. In addition, we found for the first time that CORT inhibits *Lhcgr* expression by repressing the *Nr3c1*-Creb/AP1 axis.

### 3.1. Impact of CORT on Lhcgr Expression

During the process of follicular dominance, the level of *Lhcgr* is gradually increased. At the end of the follicular phase, the surge of LH stimulates ovulation by binding to *Lhcgr* in ovarian GCs [36]. The decrease in *Lhcgr* expression in GCs causes ovulation failure due to insufficient LH stimulation [37,38]. Previously, we proved that continuous intraperitoneal injection of CORT can reduce the number of ovulations in mice [23], but it is still unclear whether CORT directly regulates the expression of *Lhcgr* in GCs. In this study, we demonstrated that continuous intraperitoneal injection of CORT into mice reduced *Lhcgr* expression in GCs collected from developing follicles, especially the preovulatory follicles, indicating that CORT might impede ovulation in females by inhibiting *Lhcgr* expression in GCs. On the other hand, studies have shown that CORT accumulation can inhibit GnRH secretion and reduce LH levels [27,28]. In addition, glucocorticoids have been reported to directly inhibit LHβ expression in gonadal cells [36,39]. It is conceivable that the downregulation of gonadotropins levels might affect GC survival and proliferation during CORT stimulation. Interestingly, our results showed that CORT treatment itself could repress GC viability. The results thus suggest that CORT-induced anovulation might also be attributed to the developmental failure of ovarian follicles caused by GC damage.

### 3.2. Lhcgr Transcriptional Regulation

The transcription of *Lhcgr* is regulated by a variety of transcription factors [40,41,42,43]. Previous studies have detected a widespread distribution of AP1 in the follicular membrane and GCs within growing follicles and corpus luteum [44]. In the ovary, gonadotropins can induce rapid and transient phosphorylation of the Creb protein in GCs, leading to the activation of the transcription of many gonadotropin-regulated ovarian genes [45]. In the case of cell damage, AP1 cannot be activated, and the phosphorylation of Creb is also inhibited, resulting in an inability to activate gene transcription [46]. Considering the fact that CORT might cause GCs damage, as mentioned previously, we asked whether AP1 and Creb are involved in CORT-regulated *Lhcgr* expression in GCs. Our in vivo and in vitro results both showed that CORT treatment reduces the levels of Creb, phosphorylated Creb (Ser133), c-Fos, and c-Jun in mouse ovarian GCs. In addition, the expression of *Lhcgr* was suppressed when GCs were treated with inhibitors of AP1 and Creb. Therefore, our data suggest that the inhibitory effects of CORT on *Lhcgr* expression might be dependent on the downregulation of AP1 and Creb in mouse ovarian GCs.

### 3.3. Effect of Nr3c1 on Lhcgr Transcription

The feedback mechanism of the hypothalamic–pituitary–adrenal axis involves the activation of glucocorticoids receptors (*Nr3c1*) by glucocorticoids [29]. After binding to corticosterone, these receptors are homodimerized or heterodimerized [47], and transferred to the nucleus to bind with other transcription factors or DNA response elements, leading to altered gene transcription [48,49,50]. However, there is no definitive evidence as to whether *Nr3c1* can specifically regulate *Lhcgr* expression in GCs under a high level of CORT. Our in vivo and in vitro results showed that CORT activated *Nr3c1* expression in ovarian GCs, which was associated with decreased *Lhcgr* expression. In contrast, inhibition of *Nr3c1* restored *Lhcgr* expression in the CORT-treated GCs. These data thus suggest that *Nr3c1* activation is required for CORT-repressed *Lhcgr* expression.

*Nr3c1* can both positively and negatively regulate gene transcription by affecting the functions of transcription factors [29,51]. Therefore, under a high level of CORT, the CORT–*Nr3c1* axis might act through certain transcription factors to regulate *Lhcgr* expression. Our results showed that the activation of *Nr3c1* reduced the protein levels of c-Fos, c-Jun, and Creb. In contrast, the *Nr3c1* inhibitor RU486 can block the CORT-induced downregulation of c-Fos, c-Jun, and Creb. These findings thus further demonstrated that CORT-mediated transcription regulation of *Lhcgr* might depend on *Nr3c1*-induced suppression of AP1 and Creb.

## 4. Materials and Methods

### 4.1. Chemicals and Antibodies

PBS was purchased from HyClone Laboratories, Inc. (Logan, UT, USA). Pregnant mare serum gonadotropin (PMSG) was purchased from Ningbo Second Hormone Factory (Ningbo, Zhejiang, China). CORT (HY-B1618), T-5224 (HY-12270), and 666-15 (HY-101120) were purchased from MCE (New Jersey, NJ, USA). Dimethyl sulfoxide (DMSO) was purchased from Sigma-Aldrich (St. Louis, MO, USA). Antibodies against *Nr3c1* and *Lhcgr* were obtained from Abclonal (Wuhan, China). Antibodies against c-Jun (9165), c-Fos (2250), Phospho-Creb (9197), and Creb (9198) were purchased from Cell Signaling Technology (Beverly, MA, USA).

### 4.2. Animals and Ethics

All the animal experiments were performed in accordance with the guidelines of the Animal Research Institute Committee at Nanjing Agricultural University. Three-to-four-week-old female ICR mice (Qing Long Shan Co., Animal Breeding Center, Nanjing, China) were housed five per cage in a temperature-controlled (22 ± 2 °C) room with a 12:12 h light-dark cycle and had ad libitum access to water and food. For the experiment involving CORT injection in vivo, 21 three-to-four-week-old mice were included in the control and CORT-injection groups. The CORT was dissolved in DMSO to reach a concentration of 0.2 mg/µL [23,52], and the control group was injected with the same volume of DMSO. Briefly presented, the treatment procedures were as follows: At 7:00 a.m. on the first day, the mice were injected with 10 IU of PMSG. The control group and the treatment group were injected with 5 µL of DMSO and 5 µL of CORT (0.2 mg/µL), respectively, every 8 h. At 24 h, 48 h, and 55 h after the PMSG injection, seven mice per group were sacrificed at each time point for the collection of GCs.

### 4.3. Cell Culture and Treatments

For the culture of the mouse primary granulosa cells, we used 80 three-to-four-week-old mice in each experiment. After intraperitoneal injection with PMSG, the mice were sacrificed and the ovaries were collected to obtain pooled GCs. These GCs were equally divided into different groups as indicated, and cultured with a medium containing DMEM/F-12 (1:1) with 10% (*v*/*v*) fetal bovine serum. The growth of the cells was observed 24 h after inoculation. When the cell plate inoculation area reached 80–90% (about 3 days after inoculation), various drug treatment experiments could consequently be performed [53,54,55]. For drug administration, the GCs were cultured in a medium containing CORT (50–300 μM) or DMSO for 12 h [56]. In some experiments, the GCs were pretreated with T5224 (75 μM) or 666-15 (7.5 μM) for 2 h, and then treated with CORT for 12 h.

### 4.4. Cell Viability Assay

Cell viability was assessed by measuring the conversion of tetrazolium salt (WST-8) to formazan according to the Cell Counting Kit-8 (CCK-8, Dojindo laboratories, Kumamoto, Japan) manufacturer’s instructions. Briefly, cells seeded in 96-well plates were grown to 90% confluency (equaling 5 × 10^4^ cells/well) following 2 days of culture. The GCs were exposed to CORT (100 μM, 200 μM, or 300 μM) or DMSO (control) for 12 h. After treatment, 10 μL of CCK-8 solution were added to each well containing 100 μL medium and incubated for an additional 3 h at 37 °C. Cell viability was then determined by reading the optical density at 450 nm using a microplate reader (Thermo Fisher Scientific, San Jose, CA, USA).

### 4.5. Western Blotting

Western blotting was performed as described previously [55]. Briefly, total protein extracts were fractioned by electrophoresis on a 4–20% Sure PAGE gel (Genscript, Nanjing, China) and transferred to PVDF membranes (Millipore, Bedford, MA, USA) by electroblotting. Nonspecific binding sites were blocked with 5% bovine serum albumin in TBST (Solarbio, Beijing, China) for 1 h. The membranes were then incubated with primary antibodies (1:1000) containing 0.25% bovine serum albumin in TBST overnight at 4 ℃. After washing in TBST three times, the membranes were incubated at room temperature for 1 h with a diluted (1:2000) secondary anti-rabbit IgG (ab6721, Abcam, Cambridge, UK). Signal detection was performed using the SuperSignal West Pico chemiluminescent substrate (Pierce, Rockford, IL, USA). The expression values of target proteins were normalized to Gapdh as the loading control.

### 4.6. Quantitative RT-PCR (qRT-PCR)

The collection of total RNA and cDNA from the GCs was performed as described previously [55]. The qRT-PCR was carried out using SYBR Premix Ex Taq (Takara, DRR420A) in a StepOnePlus™ Real-Time PCR System (Applied Biosystems, Foster City, CA, USA). The primer sequences for the target genes are listed in Appendix A. Glyceraldehyde 3-phosphate dehydrogenase (Gapdh) was used as an internal control.

### 4.7. Immunofluorescence

After the mice were injected intraperitoneally with DMSO or CORT, the ovaries were collected and fixed in buffered paraformaldehyde (4%), embedded in paraffin, sectioned to approximately 5 µm, and mounted on glass slides. the *Nr3c1* and *Lhcgr* protein levels in the GCs were detected using an immunofluorescence assay. After blocking with 5% bovine serum albumin at 37 °C for 60 min, all slides were incubated with the primary antibodies *Nr3c1* (1:100) or *Lhcgr* (1:200) overnight at 4 °C. After being washed with PBS, the ovaries were stained for 1 h with Alexa Fluor 488-conjugated goat anti-rabbit IgG (Invitrogen, A-11008, Waltham, MA, USA), and the nuclei were counterstained with DAPI. The slides were then visualized under a confocal microscope (Carl Zeiss, LSM 900, Jena, Germany).

### 4.8. RNA Interference

The siRNA specific for scrambled control and the *Nr3c1* were purchased from Gene Pharma (Shanghai, China). The siRNA transfections were performed using the Lipofectamine RNAiMAX reagent (Thermo Scientific) according to the manufacturer’s instructions. Briefly, cells seeded in 12-well plates were grown to 90% confluency following 2 days of culture. Subsequently, siRNA (80 pmol of siRNA per well) was transfected into each group, and 12 h later all groups were treated with DMSO or CORT, and then treated for 12 h. All groups were collected for further experiments. All siRNA sequences are shown in Appendix A.

### 4.9. Luciferase Reporter Gene Assay

The promoters of *Lhcgr* (−2000–0 bp), including wild type (−2000 bp) and mutated sequences, were cloned into the pGL3.0-luciferase vector and transfected by Lipofectamine 3000 (Invitrogen) according to the manufacturer’s protocols. In addition, luciferase activity was measured using the dual luciferase reporter assay system (Promega, Madison, WI, USA). For the relative luciferase activity analysis, the value of firefly luciferase was normalized to the value of Renilla luciferase.

### 4.10. Statistical Analysis

All data were presented as mean ± S.E. Analysis was performed using SPSS version 21.0 software (SPSS, Inc., Chicago, IL, USA) and GraphPad Prism version 8.0 statistical software (GraphPad, Inc., La Jolla, CA, USA). Differences between two groups were assessed using the Student’s *t*-test, and between multiple groups using one-way ANOVA. Values of *p* < 0.05 were considered significant. All experiments were repeated at least three times.

## 5. Conclusions

Taken together, our results showed a potential mechanism of CORT-induced anovulation via inhibiting *Lhcgr* expression in GCs through the CORT–*Nr3c1*–AP1/Creb signaling axis (Figure 7). Our results will help to reveal the mechanism by which CORT inhibits ovulation of animal follicles and provide a theoretical basis for probing the cause of the decline in animal reproduction performance under a high level of CORT. Therefore, developing small molecule inhibitors against this pathway might contribute to attenuating the undesirable effects of CORT on reproductive performance in humans and female animals.

## Figures and Tables

**Figure 1 ijms-23-12454-f001:**
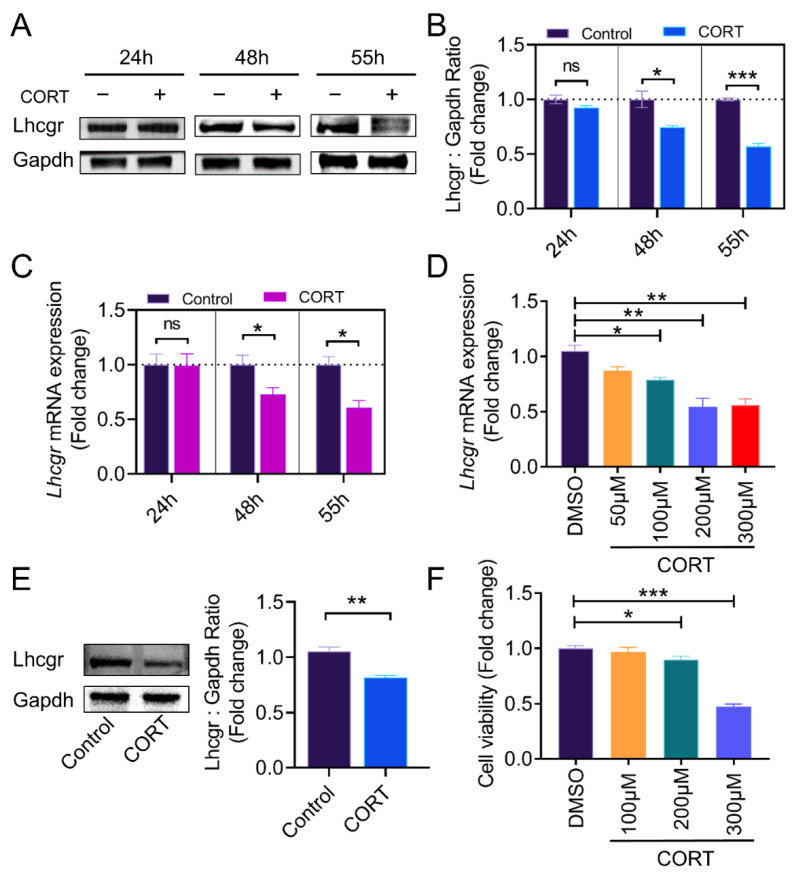
CORT (corticosterone) inhibits *Lhcgr* (LH receptor) expression. (**A**) The expression of *Lhcgr* in the ovarian GCs of mice subjected to the indicated treatment was determined at 24 h, 48 h, and 55 h after PMSG injection by western blot. (**B**) Quantification of *Lhcgr* was performed using densitometric analysis. (**C**) The *Lhcgr* in the ovarian GCs of the mice subjected to the designated treatment was measured by qRT-PCR at 24 h, 48 h, and 55 h after PMSG injection. (**D**) The qRT-PCR assay of *Lhcgr* mRNA levels in the GCs with the indicated treatments. (**E**) Primary cultured GCs were stimulated with CORT (200 μM) for 12 h. The protein levels of *Lhcgr* in the GCs were determined using western blotting, and quantification of the *Lhcgr* protein levels was performed using densitometric analysis. (**F**) Primary cultured GCs with different concentrations (100 to 300 μM) of CORT were treated for 12 h, and then the CCK-8 assay was used to determine cell viability. Data are shown as mean ± S. E.; *n* = 3. * *p* < 0.05, ** *p* < 0.01, *** *p* < 0.001, ns indicates no significant difference.

**Figure 2 ijms-23-12454-f002:**
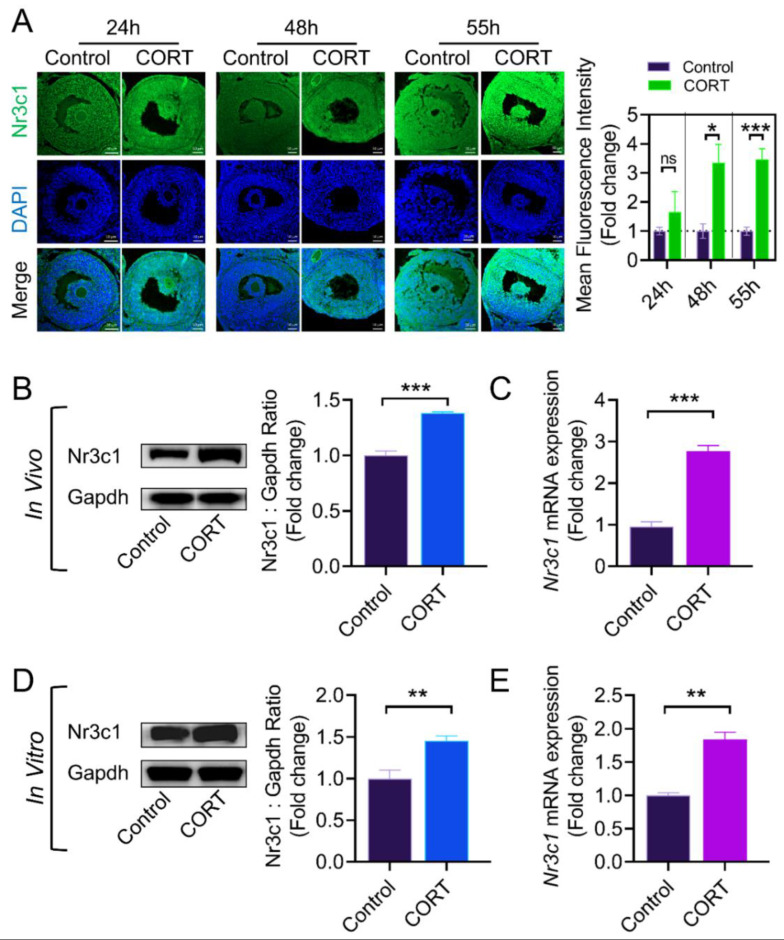
CORT activates *Nr3c1* expression. (**A**) Immunofluorescent detection of *Nr3c1* (green) in the ovarian section. The nuclei were counterstained with DAPI (blue). Scale Bar = 50 μm. (**B**) At 55 h after PMSG injection, mouse ovarian GCs were collected from the control group and the CORT injection group to detect the *Nr3c1* expression by western blot (**B**) and qRT-PCR (**C**), and the protein levels of *Nr3c1* were quantified using densitometric analysis (**B**). (**D**) Primary cultured ovarian GCs were treated with CORT (200 μM) for 12 h and then collected for western blotting (**D**) and qRT-PCR (**E**). Data are shown as mean ± S. E.; *n* = 3. * *p* < 0.05, ** *p* < 0.01, *** *p* < 0.001, ns indicates no significant difference.

**Figure 3 ijms-23-12454-f003:**
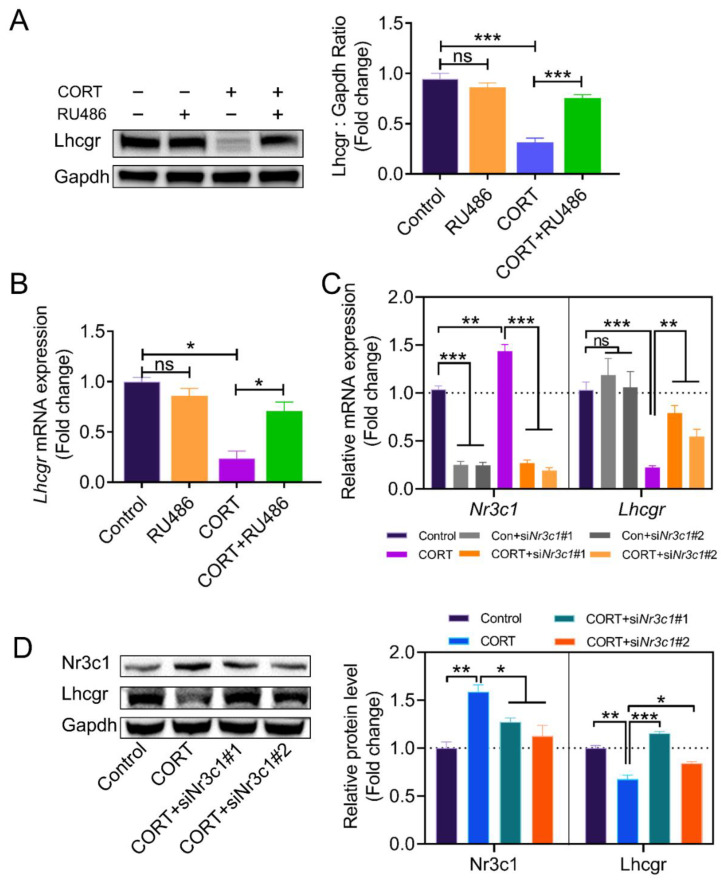
*Nr3c1* is required for the CORT-induced suppression of *Lhcgr* expression. (**A**) According to the description, the *Lhcgr* protein levels of different groups were detected by western blot, and quantification of the *Lhcgr* protein levels was performed using densitometric analysis. (**B**) The mRNA expression of *Lhcgr* in each group was detected using qRT-PCR. (**C**) Primary cultured GCs remained as an untreated control or were transfected with *Nr3c1* siRNA or scrambled control siRNA (control) for 48 h, the mRNA and protein levels of both *Nr3c1* and *Lhcgr* were detected using qRT-PCR (**C**) and western blot (**D**), and quantification of the protein levels was performed using densitometric analysis. Data are shown as mean ± S. E.; *n* = 3. * *p* < 0.05, ** *p* < 0.01, *** *p* < 0.001, ns indicates no significant difference.

**Figure 4 ijms-23-12454-f004:**
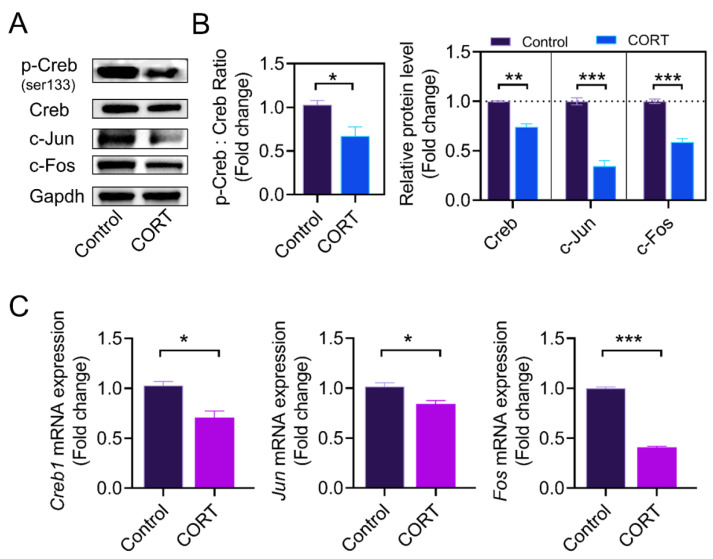
CORT injection reduces AP1 and Creb expression in vivo. (**A**,**B**) At 55 h after PMSG injection, mouse ovarian GCs were collected from the control group and the CORT injection group, the ratio of p-Creb to Creb and the levels of c-Jun and c-Fos proteins were detected using western blot, and the protein bands were quantified using densitometric analysis. Gapdh served as the control for loading. (**C**) The mRNA expressions of *Fos*, *Jun*, and *Creb1* in ovarian granulosa cells were determined by qRT-PCR. Data are shown as mean ± S. E.; *n* = 3. * *p* < 0.05, ** *p* < 0.01, *** *p* < 0.001.

**Figure 5 ijms-23-12454-f005:**
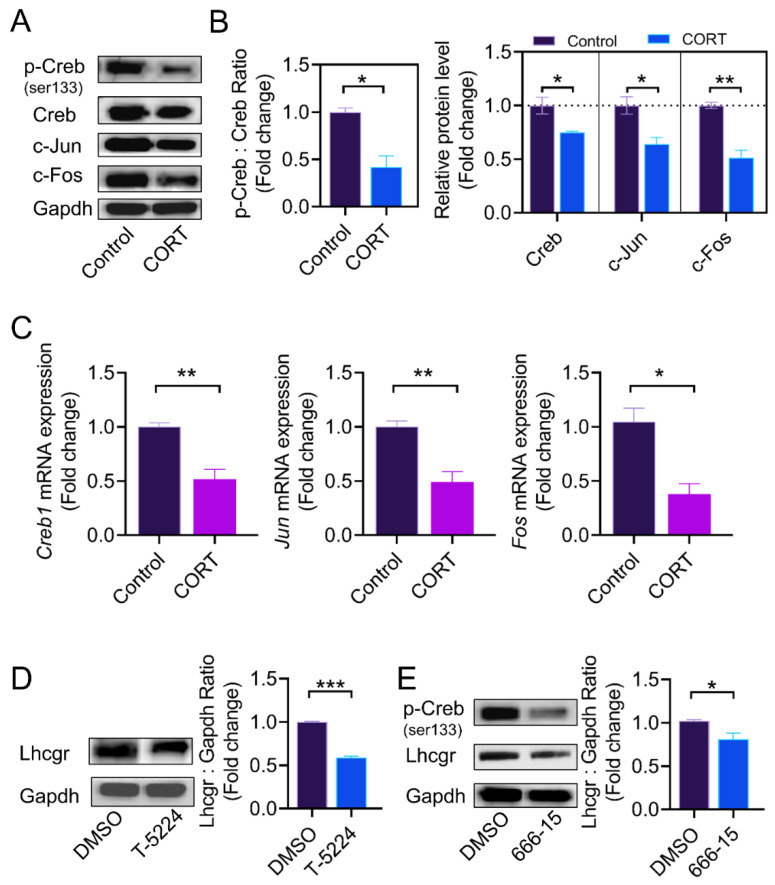
CORT inhibits the expression of AP1 and Creb in cultured GCs. (**A**,**B**) At 55 h after PMSG injection, mouse ovarian GCs were collected from the control group and the CORT injection group, the ratio of p-Creb to Creb and the levels of c-Jun and c-Fos proteins were detected using western blot, and the protein bands were quantified using densitometric analysis. Gapdh served as the control for loading. (**C**) The mRNA expressions of *Fos*, *Jun*, and *Creb1* in ovarian granulosa cells were determined using qRT-PCR. (**D**,**E**) Primary cultured ovarian GCs were treated with the AP1 inhibitor T-5224 (75 μM) and the Creb inhibitor 666-15 (7.5 μM) for 12 h, the protein levels of *Lhcgr* in the ovarian granulosa cells were determined using western blotting, and quantification of the protein levels was performed using densitometric analysis. Data are shown as mean ± S. E.; *n* = 3. * *p* < 0.05, ** *p* < 0.01, *** *p* < 0.001.

**Figure 6 ijms-23-12454-f006:**
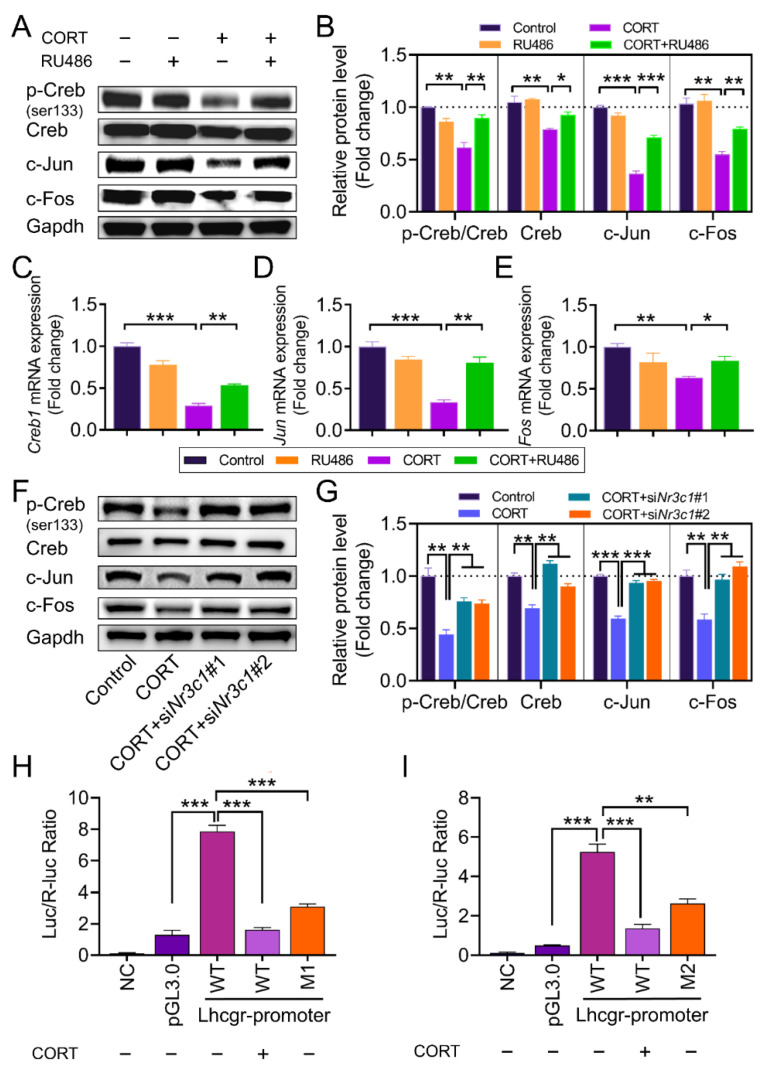
The expression of AP1 and Creb is regulated by *Nr3c1*. (**A**,**B**) Primary cultured GCs were incubated with CORT (200 μM) for 12 h. The *Nr3c1* inhibitor RU486 (50 μM) was added 2 h before the CORT treatment. The ratio of p-Creb to Creb and the protein levels of c-Jun and c-Fos were detected using western blot. The protein levels were quantified using densitometric analysis. (**C**–**E**) The mRNA expressions of *Fos*, *Jun*, and *Creb1* in ovarian granulosa cells were determined using qRT-PCR. (**F**,**G**) Primary cultured GCs remained as an untreated control or were transfected with *Nr3c1* siRNA or scrambled control siRNA (Control) for 48 h and then treated with CORT or DMSO. The ratio of p-Creb to Creb and the level of c-Jun and c-Fos were detected using western blot. Quantification of protein levels was performed using densitometric analysis (**G**). (**H**,**I**) The GCs co-transfected with *Lhcgr*-promoter luciferase—including pGL3-*Lhcgr* (WT), pGL3-*Lhcgr* (M1, mutated Creb binding site), and pGL3-*Lhcgr* (M2, mutated c-Jun binding site)—and TK-renilla luciferase vectors for 24 h were left untreated or treated with CORT for 12 h. The cells were harvested for dual luciferase assay. Data are shown as mean ± S. E.; *n* = 3. * *p* < 0.05, ** *p* < 0.01, *** *p* < 0.001.

**Figure 7 ijms-23-12454-f007:**
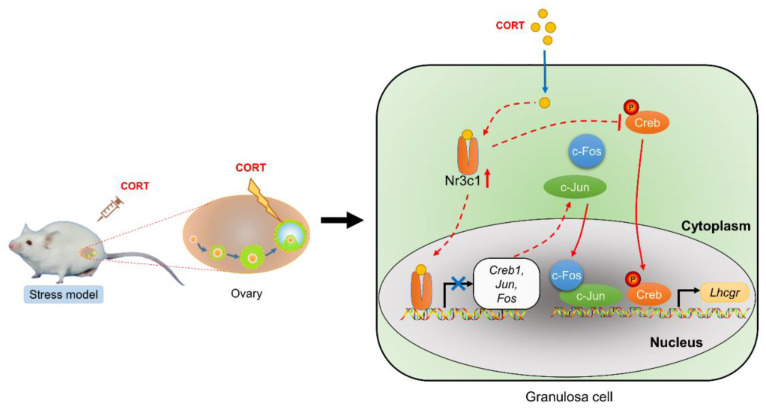
A schematic model showing the regulatory mechanism of *Lhcgr* expression in mouse GCs under corticosterone exposure. The schematic shows a potential mechanism of CORT-induced anovulation via inhibiting *Lhcgr* expression in GCs through the CORT–*Nr3c1*–AP1/Creb signaling axis. After CORT enters the granulosa cells, on the one hand, the CORT inhibits the transcription of *Lhcgr* by promoting the expression of *Nr3c1*, thereby inhibiting the transcription of Creb, Jun, and Fos, and reducing the level of the transcription factor AP-1. On the other hand, the increased level of *Nr3c1* suppresses the phosphorylation activation of Creb, which in turn suppressed the expression level of *Lhcgr*.

## Data Availability

The data underlying this article are available in the article and in its online Appendix A.

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
