# Peer review of "The Corticosterone–Glucocorticoid Receptor–AP1/CREB Axis Inhibits the Luteinizing Hormone Receptor Expression in Mouse Granulosa Cells"

_ijms, 2022, doi:10.3390/ijms232012454_

Round 1
Reviewer 1 Report
- Figure 1 legend did not match the data in figure the figure.
- Supplemental data of Western blots was unlabeled and whole blots were not shown. Poorly organized.
- Lhcgr-luciferase promoter was lacking information about the section of promtoer that was being used for experiments.
- Western blots were all over exposed and should be repeated.
Author Response
Response to Reviewer 1
1.Figure 1 legend did not match the data in figure the figure.
Response: Thank you very much for your comments. We feel sorry for the incorrect legend in figure 1, and the figure 1 legend has now been revised.
2.Supplemental data of Western blots was unlabeled and whole blots were not shown. Poorly organized.
Response: Thank you very much for your valuable comments and suggestions. We are sorry for the inappropriate blots in the Supplemental data, and we carefully reorganized the whole blots. The Western blots of Supplemental data has now been modified, and the whole blots has now been shown. For sequence information, please see the supplementary Data (Word).
3.Lhcgr-luciferase promoter was lacking information about the section of promtoer that was being used for experiments.
Response: Thank you very much for your comments. We are sorry for the unclear description in the manuscript. Actually, the information of Lhcgr-luciferase promoter was exhibited on lines 389-391.The promoter of Lhcgr (-2000 to 0 bp), including wild type and mutated sequences, were cloned into the pGL3.0-luciferase vector.
4.Western blots were all over exposed and should be repeated.
Response: Thank you for your careful reading and good suggestion. We have reduced the brightness of the picture and improved the overexposed picture and the details are in the supplementary material. The western blots showed representative bands of each protein.
Reviewer 2 Report
The manuscript "The corticosterone-glucocorticoid receptor-API/CREB axis inhibits luteinizing hormone receptor expression in mouse granulosa cells." contains interesting and novel data. However, there are several concerns with these data that need to be addressed, which are outlined below.
Major concerns:
1. I find the rationale is insufficient for utilizing the model system for all in vivo work. Mice are given PMSG to mimic pregnancy and mice are not sexually mature at only 3-4 weeks old.
2. Mice were given about 7mg of CORT via several injections within a two-day span. This seems like a lot for the size of female mice at this age. Please provide information on how this would relate to physiological levels of animals undergoing stress.
3. Figure 6H,I are missing controls.
Other concerns:
1. Figure 1 legend is incorrect, 1A in legend is not included in this figure, I believe it is the same text for figure 2A.
2. Figure 2A needs to be quantified.
3. Figure 6F- GAPDH does not seem to be a good loading control as it appears to be decreased with siRNA treatments.
4. Cell viability experiments were mentioned in methods and discussiion but I did not see them in the figures.
5. Please provide more detail for methods of siRNA knockdowns.
6. Lines 56-58 are misleading. Rodents do not make much if any cortisol due to not having 17a-hydroxylase activity in the adrenal cortex.
7. Much of the text is redundant throughout the manuscript.
8. Often times it is unclear whether authors are referring to the gene or protein since only use Lhcgr to define both. Please check nomenclature and clarify for the reader if it is unclear.
Author Response
Response to Reviewer 2
Reviewer 2: The manuscript "The corticosterone-glucocorticoid receptor-API/CREB axis inhibits luteinizing hormone receptor expression in mouse granulosa cells." contains interesting and novel data. However, there are several concerns with these data that need to be addressed, which are outlined below.
Major concerns:
- I find the rationale is insufficient for utilizing the model system for all in vivo work. Mice are given PMSG to mimic pregnancy and mice are not sexually mature at only 3-4 weeks old.
Response: We gratefully acknowledge the honorable reviewer's suggestions. We agree with you that selectively mature mice are also a great choice, but, in fact, PMSG is often injected into 3-4-week-old mice for superovulation experiment. In most cases, best results are achieved with mice ages 21-35 days (12-14 gram body weight) receiving a 2.0-5.0 IU dose of each gonadotropin.(https://cdn.shopify.com/s/files/1/0563/4487/1064/files/Ilex_Life_Sciencs_LLC_-_Mouse_Superovulation_Protocol.pdf?v=1629995449)
- Mice were given about 7mg of CORT via several injections within a two-day span. This seems like a lot for the size of female mice at this age. Please provide information on how this would relate to physiological levels of animals undergoing stress.
Response: We thank the reviewer for his/her constructive and thoughful comments. We admit that the dose of CORT used in the study is high. Indeed, this study is derived from our previous experiments (Wei et al., 2019, Animals), which have confirmed that continuous injection of CORT can be used to build a mouse stress model that negatively affect the ovulation process. The dose of CORT applicated in our experiments was based on reference from other researchers (Breen, Billings, Wagenmaker, Wessinger, & Karsch, 2005). The purpose of their research is to establish a stress animal model through repeated injections of CORT, which leads to depressive behavior. In stress-related diseases, including depression, hypothalamic-pituitary-adrenal (HPA) axis can be activated in response to stress, leading to an increase in the concentration of glucocorticoids in the circulating blood. Particularly, in intensive livestock farming, the cumulative generation of CORT in response to stress conditions is unavoidable. Using the abovementioned stress model, we aimed to investigate the role of CORT in ovulation, although this dose of CORT might cause other adverse effects on the animal. For example, at higher doses, steroid hormones can activate many classical and non-classical hormone receptors.
Wei, Y., Li, W., Meng, X., Zhang, L., Shen, M., & Liu, H. (2019). Corticosterone Injection Impairs Follicular Development, Ovulation and Steroidogenesis Capacity in Mice Ovary. Animals (Basel), 9(12). doi:10.3390/ani9121047
Breen, K. M., Billings, H. J., Wagenmaker, E. R., Wessinger, E. W., & Karsch, F. J. (2005). Endocrine basis for disruptive effects of cortisol on preovulatory events. Endocrinology, 146(4), 2107-2115. doi:10.1210/en.2004-1457
- Figure 6H,I are missing controls.
Response: Thank you very much for your comments, that is a very good question. We apologize for not expressing ourselves clearly. In figure 6 H, I,"NC" represented the control group without firefly plasmid, only Renilla luciferase plasmid was transfected, and the "pGL3.0" group was transfected with pGL3.0-luciferase vector, as empty plasmid, and Renilla plasmid, both of which existed as control in this experiment. GCs co-transfected with Lhcgr-promoter luciferase [including pGL3-Lhcgr (WT), pGL3-Lhcgr (M1, mutated Creb binding site), and pGL3-Lhcgr (M2, mutated c-Jun binding site)] and TK-renilla luciferase vectors.
Other concerns:
- Figure 1 legend is incorrect, 1A in legend is not included in this figure, I believe it is the same text for figure 2A.
Response: Thank you very much for your comments. We feel sorry for the incorrect legend in figure 1, and the figure 1 legend has now been revised.
- Figure 2A needs to be quantified.
Response: Thank you for your careful reading and good suggestion. We have supplemented the quantitative analysis of the fluorescence images, see Figure 2A.
- Figure 6F- GAPDH does not seem to be a good loading control as it appears to be decreased with siRNA treatments.
Response: Thank you for your careful reading. The question you raised is a very noteworthy point. In fact, in most experiments, the levels of Gapdh were approximately equal, such as Figure 3D. Therefore, we believe that siRNA does not lead to the reduction of Gapdh here.
- Cell viability experiments were mentioned in methods and discussiion but I did not see them in the figures.
Response: Thank you very much for your comments. Cell viability experiments were shown in figure 1F.
- Please provide more detail for methods of siRNA knockdowns.
Response: Thank you for your good suggestion. In accordance with your request, we provided more detail for methods of siRNA knockdowns in line 396.
- Lines 56-58 are misleading. Rodents do not make much if any cortisol due to not having 17a-hydroxylase activity in the adrenal cortex.
Response: Thank you for your careful reading and good suggestion. The sentence on line 55-56 has now been revised as " When rodents are exposed to stress, the level of corticosterone will increase rapidly in serum ".
- Much of the text is redundant throughout the manuscript.
Response: Thank you for your careful reading and good suggestion. We deleted or modified some redundancy sentences in the discussion to improve readability.
- Often times it is unclear whether authors are referring to the gene or protein since only use Lhcgr to define both. Please check nomenclature and clarify for the reader if it is unclear.
Response: Thank you for your careful reading and good suggestion. We have now revised the manuscript to meet the reviewer's suggestions.
Reviewer 3 Report
The authors showed that corticosterone (CORT) administration downregulates Lhcgr expression in ovarian GCs through the Nr3c1-Creb/AP1 axis. CORT activated Nr3c1 expression which was associated with a decrease in Lhcgr expression. The expression of Lhcgr was also suppressed when GCs were treated with inhibitors of AP1 (c-jun and c-fos) and Creb.
-The schematic representation of the signaling pathways involved (final figure) was a very good initiative, however, I found difficult to follow what is happening. I would recommend increasing the size of the blue cross, placing the figure after the results or discussion section, and include, in the legend, a brief explanation of the signaling pathway proposed in the study.
-Add the abbreviature of corticosterone "(CORT)" in line 16
- Pregnant mare serum gonadotropin should replace pregnant horse serum gonadotropin
- Supplementary blots should be the whole membrane (without crop) to complement the blots shown in the manuscript. Will be helpful to see all the membranes of the 3 replicates.
- The authors should explain or add references for the selected dose of corticosterone administered in mice.
- Can the authors identify the benefits/significance of this study on human reproductive care and include them in the introduction, in the first paragraph, for example, to complement the benefits for animal care? In my opinion, it would increase the impact of the study.
- Fig 1A does not correspond to a microscopy image. The (E) should be the legend of (D)?
- Microscopy images of Fig 2A seem blurry.
Author Response
Response to Reviewer 3
Reviewer 3: The authors showed that corticosterone (CORT) administration downregulates Lhcgr expression in ovarian GCs through the Nr3c1-Creb/AP1 axis. CORT activated Nr3c1 expression which was associated with a decrease in Lhcgr expression. The expression of Lhcgr was also suppressed when GCs were treated with inhibitors of AP1 (c-jun and c-fos) and Creb.
1.The schematic representation of the signaling pathways involved (final figure) was a very good initiative, however, I found difficult to follow what is happening. I would recommend increasing the size of the blue cross, placing the figure after the results or discussion section, and include, in the legend, a brief explanation of the signaling pathway proposed in the study.
Response: Thank you for your careful reading and good suggestion. We have now revised the figure to meet your suggestions. We added a brief explanation of the signaling pathway proposed in the study. In addition, figure 7 and figure 7 legend have been moved to the discussion section.
2.Add the abbreviature of corticosterone "(CORT)" in line 16.
Response: Thank you for pointing out this problem. The abbreviature of corticosterone in line 64 has now been added.
3.Pregnant mare serum gonadotropin should replace pregnant horse serum gonadotropin.
Response: We are sorry for this mistake. We sincerely apologize for our carelessness, the "horse" has now been replaced by the "mare".
4.Supplementary blots should be the whole membrane (without crop) to complement the blots shown in the manuscript. Will be helpful to see all the membranes of the 3 replicates.
Response: Thank you very much for your valuable comments and suggestions. We are sorry for the inappropriate blots in the Supplemental data, and we carefully reorganized the whole blots. The Western blots of Supplemental data has now been modified, and the whole blots has now been shown.
5.The authors should explain or add references for the selected dose of corticosterone administered in mice.
Response: Thank you very much for your comments, that is a very good suggestion. We apologize for not expressing ourselves clearly. In fact, this study is derived from our previous experiments (Wei et al., 2019), which have confirmed that continuous injection of CORT can be used to build a mouse stress model that negatively affect the ovulation process. The dose of CORT applicated in our experiments was based on reference from other researchers (Kellie et al., 2005, Endocrinology). The purpose of their research is to establish a stress animal model through repeated injections of CORT, which leads to depressive behavior. In stress-related diseases, including depression, hypothalamic-pituitary-adrenal (HPA) axis can be activated in response to stress, leading to an increase in the concentration of glucocorticoids in the circulating blood. Particularly, in intensive livestock farming, the cumulative generation of CORT in response to stress conditions is unavoidable. Using the abovementioned stress model, we aimed to investigate the role of CORT in ovulation, although this dose of CORT might cause other adverse effects on the animal.
Wei, Y., Li, W., Meng, X., Zhang, L., Shen, M., & Liu, H. (2019). Corticosterone Injection Impairs Follicular Development, Ovulation and Steroidogenesis Capacity in Mice Ovary. Animals (Basel), 9(12). doi:10.3390/ani9121047
Breen, K. M., Billings, H. J., Wagenmaker, E. R., Wessinger, E. W., & Karsch, F. J. (2005). Endocrine basis for disruptive effects of cortisol on preovulatory events. Endocrinology, 146(4), 2107-2115. doi:10.1210/en.2004-1457
- Can the authors identify the benefits/significance of this study on human reproductive care and include them in the introduction, in the first paragraph, for example, to complement the benefits for animal care? In my opinion, it would increase the impact of the study.
Response: Thank you very much for your valuable comments and suggestions. We agree that mentioning the benefits / significance of human reproductive care will increase the impact of this study, and we add the sentence to the introduction as you suggested (Line 43-45, 125-127).
7.Fig 1A does not correspond to a microscopy image. The (E) should be the legend of (D)?
Response: Thank you very much for your comments. We feel sorry for the incorrect legend in figure 1, and the figure 1 legend has now been revised.
8.Microscopy images of Fig 2A seem blurry.
Response: Thank you very much for your valuable comments and suggestions. We are sorry for the inappropriate images in Fig.2A, and we carefully reorganize and replace the images. The images has now been modified. In addition, the Mean fluorescence intensity of image was quantitatively analyzed (See Figure 2A).
Round 2
Reviewer 1 Report
The authors made suitable changes to the manuscript. However, the original images of blots are still cropped images. The journal website indicates:
"In order to ensure the integrity and scientific validity of blots (including, but not limited to, Western blots) and the reporting of gel data, original, uncropped and unadjusted images should be uploaded as Supporting Information files at the time of initial submission."
The Author's did not include uncropped and unadjusted Western blots
Author Response
The authors made suitable changes to the manuscript. However, the original images of blots are still cropped images. The journal website indicates:
"In order to ensure the integrity and scientific validity of blots (including, but not limited to, Western blots) and the reporting of gel data, original, uncropped and unadjusted images should be uploaded as Supporting Information files at the time of initial submission."
The Author's did not include uncropped and unadjusted Western blots
Response: Thank you very much for your comments. Due to operational errors, we did not successfully upload the reply file to you, and we are very sorry for this. We are sorry for the inappropriate original images in the manuscript. Due to the data loss of a previous researcher, some of the original blots could not be obtained, so we were in a race against time to re-conduct this part of the experiment to replace and improve the inappropriate blots. We hope you can understand. We have uploaded the new results, please check the new manuscript and original blots (Included in Supplementary Data). Thank you again for your reminder and reasonable suggestions, which will be very helpful for us.
Reviewer 2 Report
Thank you for kindly addressing my previous points. After reviewing the citation you provided regarding the protocol for super ovulation, it appears you did not exactly follow that protocol. I am not an expert in this area and will leave this concern up to the discretion of the editor.
Please see below for existing comments/concerns.
Lines 106-110 contain redundant information, please modify for concision.
Please provide the concentration of siRNA used for knockdown experiments.
I find the controls for some of the experiments are still lacking (listed below):
Figure 3, there is no protein expression for siNr3c1 alone, only shown in the presence of CORT and the comparison with WT and knockdown in the presence of CORT although significant is less than 50%.
Figure 6, as mentioned in your response the GAPDH in previous experiments is relatively unchanged, so the reduced GAPDH in knockdown may be a result of reduced protein loading, which is also a concern.
Western blots in general are overexposed, even in the revised supplement, many of these blots are not of publication quality. Some look pixelated, while others are cut off (figure 6F-CREB), lanes run together (figure 4-cFos), and very high background, too high to quantify band (figure 6F, cFOS control). Perhaps these are not the best choice for representative images across the experiments.
As mentioned previously, my expertise is not in pregnancy/fertility so I am not sure of the literature in this area. However, often times CORT treatment reduces Nr3c1 expression due to negative feedback. Please comment on potential reasons why you are observing the opposite.
Lastly, small point but generally the nomenclature for the glucocorticoid receptor is Nr3c1 for gene expression, and glucocorticoid receptor (or GR) when referring to the protein.
Author Response
Response to Reviewer 2
Thank you for kindly addressing my previous points. After reviewing the citation you provided regarding the protocol for super ovulation, it appears you did not exactly follow that protocol. I am not an expert in this area and will leave this concern up to the discretion of the editor.
Response: Thank you for your comments, this is indeed a matter of concern. Mouse ovaries have important endocrine functions that synthesize and release steroid hormones, such as estrogen and progesterone, which can influence the development of the ovaries themselves, resulting in some variation in the different follicular developmental stages. In sexually mature mice, the follicular phase can be divided into follicular and luteal phases. The follicular phase refers to the process from the beginning of follicular development to maturation and ovulation, a process that is mainly characterized by an increase in estrogen levels. The luteal phase refers to the process from the beginning of the formation of the corpus luteum to its disappearance, during which progesterone levels are elevated. These hormonal changes lead to different stages of follicular development in the ovaries of mice, which is not conducive to the study of ovulation mechanisms. In contrast, in sexually immature mice, none of the follicles in the ovary are activated, and after exogenous PMSG injection, they simultaneously activate development and activate ovulation signals. Therefore, 3-4 weeks old unsexually mature mice are usually selected for ovulation mechanism studies1.
Please see below for existing comments/concerns.
- Lines 106-110 contain redundant information, please modify for concision.
Response: Thank you for your careful reading and good suggestion. We feel sorry for the redundant sentences, and the sentences of Lines 106-110 have now been revised.
- Please provide the concentration of siRNA used for knockdown experiments.
Response: Thank you for your comments. According to the manufacturer's instructions, in this knockdown experiment, we added an amount of 80 pmol of siRNA per well in a 12-well plate, and the knockdown efficiency was validated at the transcriptional level (Fig 3C) in the new manuscript.
I find the controls for some of the experiments are still lacking (listed below):
- Figure 3, there is no protein expression for siNr3c1 alone, only shown in the presence of CORT and the comparison with WT and knockdown in the presence of CORT although significant is less than 50%.
Response: Thank you for your comments. As mentioned in point 2, the knockdown efficiency was validated at the transcriptional level (Fig 3C). Firstly, gene expression is divided into two processes: transcription and translation. Gene expression is divided into two processes: transcription and translation, which are expressed at the mRNA level and protein level, and differences in the timing and location of transcription and translation of eukaryotic genes. Secondly, after transcription, a series of processes such as post-transcriptional processing, translation, and post-translational modification will take place. Therefore, it is not completely consistent at the transcriptional level and the translation level.
- Figure 6, as mentioned in your response the GAPDH in previous experiments is relatively unchanged, so the reduced GAPDH in knockdown may be a result of reduced protein loading, which is also a concern.
Response: Thank you for your good suggestion. To confirm this result, we re-conduct this part of the experiment to replace and improve the inappropriate blots.
- Western blots in general are overexposed, even in the revised supplement, many of these blots are not of publication quality. Some look pixelated, while others are cut off (figure 6F-CREB), lanes run together (figure 4-cFos), and very high background, too high to quantify band (figure 6F, cFOS control). Perhaps these are not the best choice for representative images across the experiments.
Response: Thank you very much for your comments. We are sorry for the inappropriate original images in the manuscript. Due to the data loss of a previous researcher, some of the original blots could not be obtained, so we were in a race against time to re-conduct this part of the experiment to replace and improve the inappropriate blots. We hope you can understand. We have uploaded the new results, please check the new manuscript and original blots. Thank you again for your reminder and reasonable suggestions, which will be very helpful for us.
- As mentioned previously, my expertise is not in pregnancy/fertility so I am not sure of the literature in this area. However, often times CORT treatment reduces Nr3c1 expression due to negative feedback. Please comment on potential reasons why you are observing the opposite.
Response: Thank you very much for your comments. Indeed, several studies have shown that an increase in corticosterone concentration would lead to an inhibition of Nr3c1 expression2, 3. However, in contrast, many other studies have found that corticosterone positively correlates with Nr3c1expression or that corticosterone directly promotes Nr3c1 expression in a variety of tissues and cells4-7. Therefore, we suggest that corticosterone may have different regulatory effects on the transcriptional level of Nr3c1 in different tissues or cells. In addition, the dose of corticosterone injection or treatment may alter the expression of Nr3c1.
- Lastly, small point but generally the nomenclature for the glucocorticoid receptor is Nr3c1 for gene expression, and glucocorticoid receptor (or GR) when referring to the protein.
Response: Thank you for your careful reading and good suggestion. Nr3c1 is also a commonly used protein symbol and has been used in many pieces of literature to represent protein names8-11. Therefore, we do not think it is inappropriate to use Nr3c1 here. Thank you again for your comments.
Reference
(1) Rey, R.; Josso, N.; Racine, C. Sexual Differentiation. In Endotext, Feingold, K. R., Anawalt, B., Boyce, A., Chrousos, G., de Herder, W. W., Dhatariya, K., Dungan, K., Hershman, J. M., Hofland, J., Kalra, S., et al. Eds.; 2000.
(2) Marelli, S. P.; Terova, G.; Cozzi, M. C.; Lasagna, E.; Sarti, F. M.; Cavalchini, L. G. Gene expression of hepatic glucocorticoid receptor NR3C1 and correlation with plasmatic corticosterone in Italian chickens. Anim Biotechnol 2010, 21 (2), 140-148.
(3) Yuan, H. J.; Han, X.; He, N.; Wang, G. L.; Gong, S.; Lin, J.; Gao, M.; Tan, J. H. Glucocorticoids impair oocyte developmental potential by triggering apoptosis of ovarian cells via activating the Fas system. Sci Rep 2016, 6, 24036.
(4) Feng, Y.; Li, Y.; Jiang, W.; Hu, Y.; Jia, Y.; Zhao, R. GR-mediated transcriptional regulation of m(6)A metabolic genes contributes to diet-induced fatty liver in hens. J Anim Sci Biotechnol 2021, 12 (1), 117.
(5) Santos, T. B.; Cespedes, I. C.; Viana, M. B. Chronic corticosterone administration facilitates aversive memory retrieval and increases GR/NOS immunoreactivity. Behav Brain Res 2014, 267, 46-54.
(6) Tesic, V.; Ciric, J.; Jovanovic Macura, I.; Zogovic, N.; Milanovic, D.; Kanazir, S.; Perovic, M. Corticosterone and Glucocorticoid Receptor in the Cortex of Rats during Aging-The Effects of Long-Term Food Restriction. Nutrients 2021, 13 (12).
(7) Vojnovic Milutinovic, D.; Teofilovic, A.; Velickovic, N.; Brkljacic, J.; Jelaca, S.; Djordjevic, A.; Macut, D. Glucocorticoid signaling and lipid metabolism disturbances in the liver of rats treated with 5alpha-dihydrotestosterone in an animal model of polycystic ovary syndrome. Endocrine 2021, 72 (2), 562-572.
(8) Li, X. M.; Li, M. T.; Jiang, N.; Si, Y. C.; Zhu, M. M.; Wu, Q. Y.; Shi, D. C.; Shi, H.; Luo, Q.; Yu, B. Network Pharmacology-Based Approach to Investigate the Molecular Targets of Sinomenine for Treating Breast Cancer. Cancer Manag Res 2021, 13, 1189-1204.
(9) Piovan, E.; Yu, J.; Tosello, V.; Herranz, D.; Ambesi-Impiombato, A.; Da Silva, A. C.; Sanchez-Martin, M.; Perez-Garcia, A.; Rigo, I.; Castillo, M.; et al. Direct reversal of glucocorticoid resistance by AKT inhibition in acute lymphoblastic leukemia. Cancer Cell 2013, 24 (6), 766-776.
(10) Real, P. J.; Tosello, V.; Palomero, T.; Castillo, M.; Hernando, E.; de Stanchina, E.; Sulis, M. L.; Barnes, K.; Sawai, C.; Homminga, I.; et al. Gamma-secretase inhibitors reverse glucocorticoid resistance in T cell acute lymphoblastic leukemia. Nat Med 2009, 15 (1), 50-58.
(11) Xiao, H.; Ding, Y.; Gao, Y.; Wang, L. M.; Wang, H.; Ding, L.; Li, X.; Yu, X.; Huang, H. Haploinsufficiency of NR3C1 drives glucocorticoid resistance in adult acute lymphoblastic leukemia cells by down-regulating the mitochondrial apoptosis axis, and is sensitive to Bcl-2 blockage. Cancer Cell Int 2019, 19, 218.
Round 3
Reviewer 2 Report
I appreciate the authors thorough responses and appropriate edits to the manuscript. I find this version to be much improved.
Please include the concentration of siRNA that you provided in the response within the methods section of the manuscript. I find this information to be helpful to fellow scientists.
Thank you.
Author Response
Thank you very much for your careful reading and comments, we gratefully acknowledge the honorable your suggestions. We have provided the concentration of siRNA in the methods section (Line 403) of the manuscript.